# HDXmodeller: an online webserver for high-resolution HDX-MS with auto-validation

Ramin Ekhteiari Salmas[1] & Antoni James Borysik[1✉]

The extent to which proteins are protected from hydrogen deuterium exchange (HDX) provides valuable insight into their folding, dynamics and interactions. Characterised by mass spectrometry (MS), HDX benefits from negligible mass restrictions and exceptional throughput and sensitivity but at the expense of resolution. Exchange mechanisms which naturally transpire for individual residues cannot be accurately located or understood because amino acids are characterised in differently sized groups depending on the extent of proteolytic digestion. Here we report HDXmodeller, the world's first online webserver for high-resolution HDX-MS. HDXmodeller accepts low-resolution HDX-MS input data and returns high-resolution exchange rates quantified for each residue. Crucially, HDXmodeller also returns a set of unique statistics that can correctly validate exchange rate models to an accuracy of 99%. Remarkably, these statistics are derived without any prior knowledge of the individual exchange rates and facilitate unparallel user confidence and the capacity to evaluate different data optimisation strategies.

[1] Department of Chemistry, Britannia House, King's College London, SE1 1DB London, UK. ✉email: antoni.borysik@kcl.ac.uk

Hydrogen deuterium exchange mass spectrometry (HDX-MS) is a biophysical technique that probes time-dependent mass changes in proteins arising from the spontaneous exchange of labile protons for deuterium in $D_2O$ solvent[1,2]. Information on the kinetics of isotope exchange can reveal important information on protein dynamics, structure, and interactions[3–6]. Recent commercialisation has facilitated greater accessibility of HDX-MS to non-specialists contributing to an upsurge in popularity of the technique. HDX-MS benefits from high sensitivity and throughput and this, coupled to an exceptional mass range and a tolerance for background contaminants such as lipids, has helped HDX-MS become established as an enabling method to investigate challenging protein systems of high biological importance[7–9]. A significant limitation of HDX-MS is its poor resolution and while HDX occurs for every amino acid except proline, exchange rates are evaluated by MS as time-dependent mass shifts in proteolytically cleaved peptides. Isotope uptake cannot be pinpointed to individual residues and important metrics of protein stability and folding, such as HDX protection factors (PFs), cannot be determined. HDX-MS is limited to asking qualitative questions about changes in protein behaviour such as those arising from point mutations or binding. While this information can provide significant insight into protein function, the utility of HDX-MS would be extended significantly if exchange rates could be characterised for each residue.

Extracting residue resolved exchange rates from experimental HDX-MS data requires some form of exchange rate modelling[10–15]. Unfortunately, modelling HDX-MS data has proven challenging and in the few cases where systematic validation has been provided, the accuracy of modelled outputs have not been encouraging. Underdetermination is the main limitation as the variables typically greatly outnumber the constraints such that many different microscopic exchange rate models are equally consistent with an experimental profile. A potential remedy for this limitation is to use additional restraints encoded by the peptide ion envelopes which can reveal clues regarding the distribution of isotope along a peptide[16–18]. However, the interpretation of ion spectra can be challenging and any changes in peak shapes arising from extraneous isotope exchange may be impossible to properly account for. A related and often overlooked problem with modelling HDX-MS data relates to insufficient understanding of what constitutes appropriate input data for modelling, beyond an acceptance that a high peptide redundancy is preferable. Differences in the success of exchange rate modelling should be anticipated for different datasets, but it is currently impossible to deduce the utility of any given input file or evaluate the accuracy of a model output without prior knowledge of the residue resolved rates. An acceptance of the challenges associated with modelling HDX-MS data, combined with an inability to validate exchange rate models, has resulted in scepticism towards these approaches. High-resolution HDX-MS would represent a significant breakthrough in the field but at present this challenge remains unresolved.

Here we report HDXmodeller the world's first online webserver for high-resolution HDX-MS. HDXmodeller is a fully automated advanced programming tool capable of calculating residue resolved HDX protection factors (PFs) from peptide level HDX-MS input data. Through an extensive search of different algorithms and procedures, HDXmodeller is able to provide the most accurate high-resolution exchange rate models currently reported, depending on the input. The standout feature of HDXmodeller however, is an auto-validation function that takes into consideration the quality of the entire optimisation process through the use of a novel method based on a covariance matrix over different replicates. Crucially, the auto-validation feature can quantify the fidelity of the model output to an accuracy of 99% without prior knowledge of the underlying residue exchange rates. HDXmodeller will provide the growing number of HDX-MS practitioners easy access to high-resolution HDX-MS along with essential insight into the reliability of their data.

## Results

**The optimisation method**. The time-dependent mass shifts of proteolytically cleaved peptides, commonly reported as the ratio of the observed mass change to the total possible mass change or relative fractional uptake (RFU), provide a potential framework for exchange rate modelling because multiple peptides typically sample the same amino acids[19]. Global optimisation of HDX-MS data should, therefore, be feasible but will depend on poorly understood parameters relating to the quality of the input data. A bottom-up optimisation strategy was utilised to develop HDXmodeller wherein HDX-MS data were built using experimental protein-peptide maps onto which RFU were projected using predefined exchange rates ($k_{obs}$) for each residue. This allowed the preparation of reference HDX-MS data for which the underlying $k_{obs}$ of each amino acid was known, such that the accuracy of HDXmodeller could be evaluated fully for every residue within a dataset across a wide range of exchange rates. Simulated datasets were used to be certain that all reference data are error-free thereby providing an unambiguous benchmark based on currently accepted HDX theory. To ensure that simulated datasets best maintained the overall character of HDX all lnP values were simulated from protein structures using well-known expressions of protein HDX behaviour. (Methods, Supplementary Fig. 1).

HDXmodeller is based on constrained nonlinear optimisation and utilises sequential quadratic programming (SQP) to solve an objective function defined by the error between the model and input RFU across all timepoints. Minimisation is initiated with random initial guess data for $k_{obs}$ which are then optimised sequentially with respect to the objective function. Multiple replicate runs are made each with different initial guess values for $k_{obs}$ after which the data from all replicates are combined. In all cases $k_{obs}$ is expressed as the natural log of the PF (lnP) which considers the chemical exchange rate ($k_{ch}$) of each residue, Eq. (1) (Methods)[20,21].

$$PF = \frac{k_{ch}}{k_{obs}} \qquad (1)$$

Critical to the success of HDXmodeller is an advanced algorithm which manages the optimisation process, and which allows automatic handling of the constraints over the objective function in every iteration. During the development of HDXmodeller, many different reference datasets were tested across different peptide maps and varying RFU timepoints. Following optimisation, the performance of HDXmodeller was remarkable with $R^2$ between model and reference lnP >0.9 in some instances and with >80% of the projected values within ±1 lnP of reference data.

The ability of HDXmodeller to accurately calculate lnP values varied throughout different HDX-MS datasets (Fig. 1a, b). Changes in the fidelity of modelled lnP was anticipated, however, and presumably reflects variations in redundancy, which reports the number of different peptides that are occupied by each residue. High redundancy is considered important for good optimisation outcomes as it should constrain the range of available $k_{obs}$ increasing precision across multiple optimisation replicates. There is also potential for the variation introduced from poorly constrained regions to propagate into the amino acids of neighbouring peptides, such that the elimination of

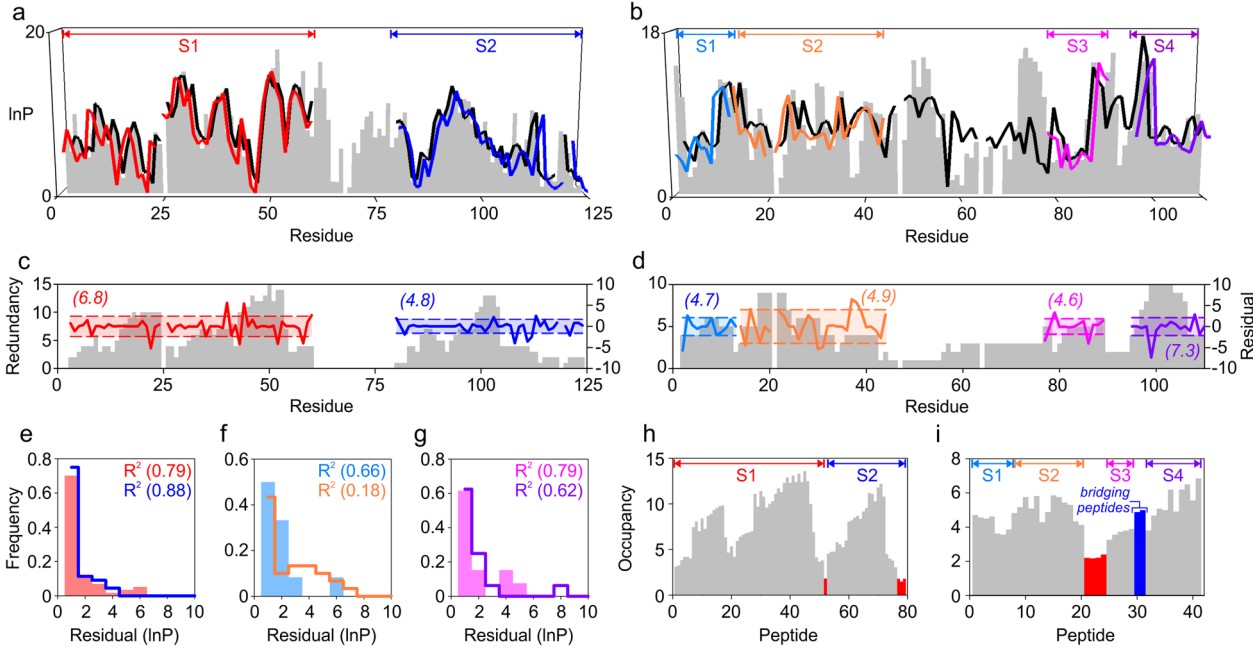

**Fig. 1 Example optimisation outputs for HDXmodeller. a, b** Comparison of reference and model lnP values built onto experimental peptide maps obtained for alpha lactalbumin (**a**) and barnase (**b**). Reference lnP values are shown as grey bars and model lnP are also shown for the whole submission (black lines) or subsections (coloured lines) as the median values across all optimisation replications. Different subsection submissions are denoted (S), breaks in the plots occur from missing data or from proline residues. **c, d** Comparison of the residue redundancy and lnP residuals of alpha lactalbumin (**c**) and barnase (**d**). Redundancy is shown as grey bars (right axis scale) and the residual lnP are shown as coloured lines as displayed in (**a, b**) (left axis scale). Shaded area represents the RMSE of the residuals for each subsection and the average redundancy for each subsection is also given in parenthesis (*n* = 100). **e–g** Residual histograms for data shown in (**a, b**) with data colouring matching the respective subsections. The R² between the reference lnP and the median model lnP are also shown in parenthesis. **h, i** Peptide occupancy for alpha lactalbumin (**h**) and barnase (**i**) data shown in (**a, b**). Peptides for different subsections are denoted (S), red bars indicate peptides with an occupancy threshold below the 2.5 cut off, blue bars denote bridging peptides that were deleted to create subsections S3 and S4 (Supplementary Fig. 2).

peptides that comprise these regions may be preferable. To account for this a quality control measure was introduced to score peptides based on their occupancy, which is the sum of residue redundancies for any peptide, or density, divided by the number of residues excluding proline and the amino-terminus (Eq. (2)), where $D$ and $n$ represent the density and total number of amino acids numbers, respectively.

$$Occupancy = \frac{1}{n} \sum_{k=1}^{n} D_k \qquad (2)$$

An occupancy threshold of 2.5 was deemed optimal for the identification of weak peptides that should be excluded prior to analysis (Fig. 1h–i, Supplementary Fig. 2). Where possible, HDX-MS data were also split into different subsections and each subsection optimised independently.

Because the accuracy of HDXmodeller naturally varies in response to changes in the quality of input data, subdivided inputs provide more scope for guiding potential users on anticipated modelling outcomes at a local level. However, the ability to provide this information requires a greater understanding of the relationship between input data and the accuracy of the model outputs. No significant deterioration in the performance of HDXmodeller was observed when protein data was optimised by subsection and in many cases, this improved the accuracy of the model lnP (Fig. 1a–d).

We next investigated different HDX-MS datasets to understand aspects that had the greatest impact on the optimisation. Given the dependence between model accuracy and input data, guidance regarding anticipated optimisation outcomes is essential for user confidence. Contrary to our expectations, redundancy is a poor

guide of HDX-MS data quality. Although it is apparent that the overall redundancy should exceed a certain threshold it has little additional bearing on the quality of model outputs. HDX-MS datasets with the same overall redundancy scores can yield markedly different errors in lnP estimations and data with exceptionally high redundancy can perform significantly less well than data for which the redundancy score is low (Fig. 1c–g). We developed our own metrics to score input data that were based on redundancy, but which also considered the overall peptide distribution. Many different methods were developed and tested but predicting the quality of modelling outcomes from the HDX-MS peptide maps was not possible. To understand this further three different RFU datasets were prepared and projected onto an identical peptide map prior to optimisation by HDXmodeller. Despite these data being built from the same peptides there were large differences in the capacity of HDXmodeller to accurately model the lnP (Supplementary Fig. 3). This indicates that modelling outcomes cannot be predicted at the peptide level alone and have a strong additional dependence on the RFU. This is a problem because RFU values cannot be easily separated from the underlying $k_{obs}$ of each residue on which they depend. The capacity to successfully gauge modelling outcomes deductively from HDX-MS data may not, therefore, be possible as it could rely on knowledge of the microscopic exchange rates for which HDX-MS cannot provide direct access.

**The auto-validation method.** Although it was not possible to predict the quality of modelling outcomes directly from input data, gauging the accuracy of modelled lnP post-optimisation may be feasible. To achieve this, we developed a novel auto-validation matrix that calculates the pair-wise correlation

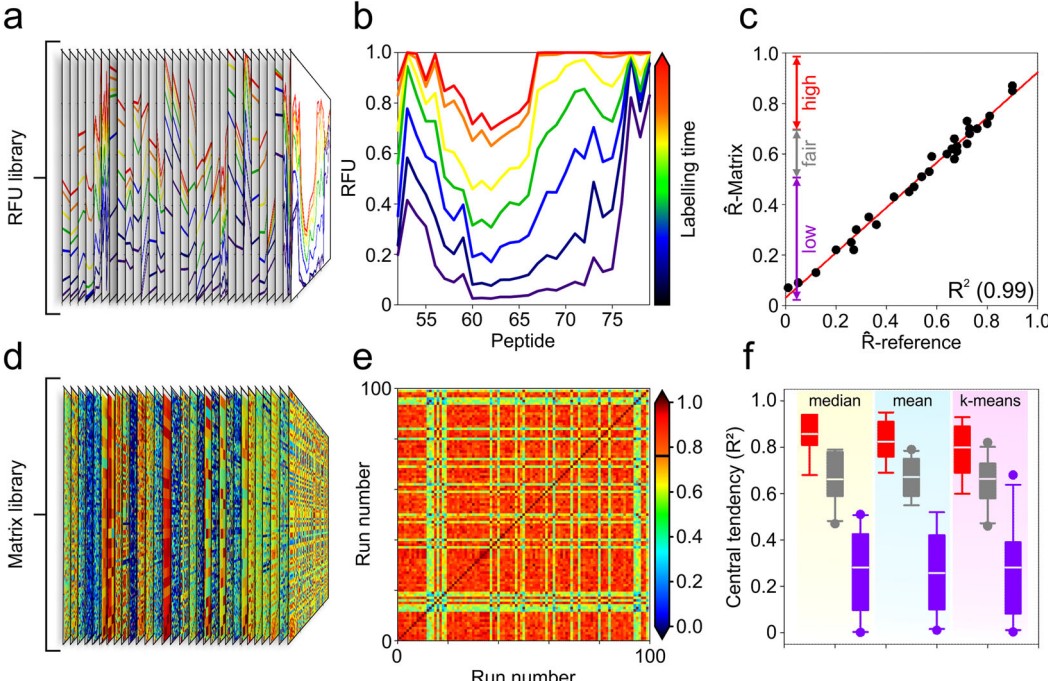

**Fig. 2 Validating the accuracy of model lnP for data subsections. a, b** Library of 30 different HDX-MS datasets used to validate R̂-matrix values obtained from the auto-validation matrices. An example reference dataset is shown displaying the RFU of different peptides at 7 different labelling times from 15 s (indigo) to 8 h (red). **c** Relationship between R̂-reference and R̂-matrix calculated following optimisation of the HDX-MS reference library. Three different classifications are shown depending on whether the accuracy of modelled lnP values are predicted to be high (>0.7), fair (0.5–0.69), or low (<0.5). **d, e** Library of auto-validation matrices for reference data depicted (**a, b**). An example matrix is shown for an optimisation performed with 100 replicates (n = 100) with each datapoint colour reporting the correlation coefficient between any 2 runs. The scalebar to the right of the plot reports the correlation coefficients of respective datapoints with the dark line across the scalebar denoting R̂-matrix, which is obtained from the average or all pairwise calculations. **f** Boxplot showing the distribution of $R^2$ between modelled and reference lnP calculated for each reference dataset. The plot shows the effect of different central tendencies, median, mean and k-means clustering on the $R^2$ with the data binned into three different classifications as shown (**c**). Plot demonstrates the superior accuracy of the median for reporting the lnP of replicate optimisation runs along with an improved capacity to maintain separation between the high, fair, and low accuracy classifications.

coefficients for every replicate over the course of a whole optimisation run. A library of reference HDX-MS data was prepared comprising of 30 different input files encompassing a total of over 500 peptides with a broad range of different sizes, redundancies, peptide distributions, RFU and lnP (Fig. 2a, b). Each dataset was submitted for optimisation by HDXmodeller and an auto-validation matrix prepared for each output from which the mean correlation coefficient (R̂-matrix) was obtained (Fig. 2d, e). The mean correlation coefficients were then recalculated for each replicate run but with the values obtained using the reference lnP values (R̂-reference) rather than pairwise between replicates (Methods). The R̂-matrix and R̂-reference scores were then compared across all datasets to see if the accuracy of the optimisation could be predicted from the matrices. On comparison, the ability of the auto-validation matrices to predict the accuracy of the optimisation was outstanding with a $R^2$ value of 0.99 (Fig. 2c). We also compared the accuracy of different central tendencies and clustering approaches including the mean, median, and k-means clustering. For data bins with R̂-matrix values of 1.0–0.7, 0.5–0.69, and 0–0.49, the median was the best performing with regard to overall accuracy and ability to distinguish between the different R̂-matrix bin classifications (Fig. 2f). Without prior knowledge of the underlying exchange rates the R̂-matrix score can provide highly accurate information on the fidelity of modelled outputs allowing HDXmodeller to assign confidence to the models. Furthermore, since the matrices operate post-optimisation they have the additional advantage of

providing potential users with the flexibility to test and score different optimisation strategies for their data.

The workflow for HDXmodeller entails data optimisation following calculation of R̂-matrix values to ascertain the accuracy of the modelled lnP. Data should also be submitted in subsections to provide more local guidance on the fidelity of the outputs. However, even poorly restrained input data, that yield low R̂-matrix values, can contain many highly accurate model lnP accounting for 50% of the residues with values that are within ±1.0 lnP of the true value. Conversely, many datasets with strong optimisation outcomes contain a small fraction of outlying data with lnP values >2.0. It would be useful therefore, if the capabilities of HDXmodeller could be extended so that it was capable of capturing individual residues with highly accurate lnP or was able to highlight outliers. We prepared optimisation histograms for each residue which report the density of lnP values over all replicates. These histograms serve as the primary guide for the accuracy of each residue and in-house benchmarking has shown that a high degree of confidence should be ascribed to residues that produce unimodal histograms with narrow distributions.

However, most residues with accurate lnP yield more varied histograms from which the accuracy of the output is more difficult to assign. This is due to the requirement to report a central tendency which can occupy a range of distances from the true value depending on the details of each optimisation (Fig. 3a, b). We tested a range of metrics to report the accuracy of the model

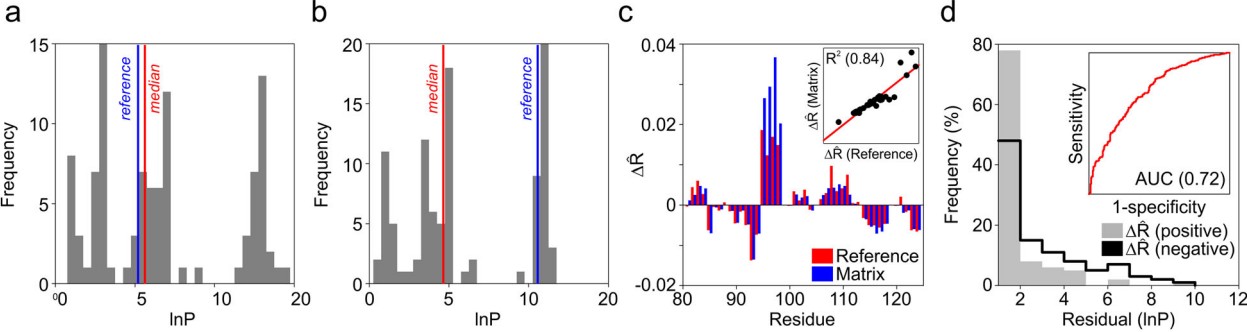

**Fig. 3 Validating the accuracy of model lnP for residues. a, b** Example optimisation histograms for two residues with high (**a**) and low (**b**) accuracy. The grey bars show the frequency of different lnP bins for an optimisation performed with 100 replicates ($n = 100$) along with the reference and median lnP reported as blue and red lines, respectively. **c** Bar chart showing the relationship between $\Delta\hat{R}$-reference (red) and $\Delta\hat{R}$-matrix (blue) values for each residue over a single HDX-MS reference dataset. The $\Delta\hat{R}$-matrix value of each residue reports the change in $\hat{R}$-matrix following its elimination from the dataset with $\Delta\hat{R}$-reference referring to changes in $\hat{R}$-reference following deletion. Plot shows the equivalence between $\Delta\hat{R}$-matrix and $\Delta\hat{R}$-reference with an $R^2$ value of 0.84 (insert). **d** Histogram of residual lnP for the modelled data separated into positive (grey bars) and negative (black line) $\Delta\hat{R}$-matrix values. Plot demonstrates 60% enrichment of highly accurate modelled values, within $\pm 1.0$ lnP of reference data, for residues with a positive $\Delta\hat{R}$-matrix. Conversely, residues with negative $\Delta\hat{R}$-matrix values are 300% more likely to have outlying lnP >2.0 from reference data. The inset shows the associated receiver operating characteristic (ROC) plot demonstrating the ability of the error in the modelled lnP to classify residues with positive and negative $\Delta\hat{R}$-matrix values. The accuracy of the classification is 72% taken from the area under the curve (AUC). All plots in (**d**) report the combined trend for all residues over the whole reference library of 30 different HDX-MS datasets ($n = 30$).

lnP for each residue, but none were successful. Ultimately, we returned to the auto-validation matrices and used them to report the change in the $\hat{R}$-matrix value ($\Delta\hat{R}$-matrix) for a dataset following the sequential removal of each residue. A positive $\Delta\hat{R}$ value should result following the deletion of residues with low error and the opposite should be true for less accurate amino acids. Equivalent calculations were then made but with $\Delta\hat{R}$ calculated against the reference lnP (Methods). In most datasets the trend in $\Delta\hat{R}$-reference was mirrored by $\Delta\hat{R}$-matrix showing that information regarding the accuracy of each residue could be predicted from the $\Delta\hat{R}$-matrix score (Fig. 3c). We then calculated the direction of $\Delta\hat{R}$-matrix for all residues in the HDX-MS reference library and investigated the capacity of this metric to rank individual residues by their error (Methods). The results indicated that the error in the model lnP of each residue could assign positive and negative $\Delta\hat{R}$-matrix values with an accuracy of 72%. Over the whole HDX-MS reference library residues with positive $\Delta\hat{R}$-matrix values contained 60% more highly accurate lnP with RMSE < 1.0 lnP. Conversely, outlier lnP values with errors >2.0 lnP were approximately three times more likely to occur in residues with a negative $\Delta\hat{R}$-matrix (Fig. 3d). Gauging the accuracy of modelled data at residue resolution is extremely challenging and no readymade method can be successfully applied to this problem. Nevertheless, the $\Delta\hat{R}$-matrix value of a residue can provide important clues regarding model accuracy and confidence to the lnP of individual amino acids.

## Discussion

HDXmodeller is available at https://hdxsite.nms.kcl.ac.uk/. The website provides guidance on data submission and processing along with all of the reference datasets used for code development including all the simulated lnP values and their associated peptide/RFU data. Users should upload their HDX-MS data as text files reporting the RFU of each isotope labelling time. It is critical that users correct their HDX-MS data for extraneous exchange prior to optimisation. Results obtained from data where the RFU have not been corrected for back and forward exchange will produce unreliable results and we recommend the method of Zhang for data correction[1]. After users have uploaded their input files they are sent to a production area for optimisation and the

outputs returned to users by email once the calculations are complete.

HDXmodeller was specifically developed with a recognition of the innate variations in the quality of input data, with regard to differences in the restraints and to our knowledge is the first HDX-MS optimisation method capable of reporting on model fidelity. The exceptional accuracy with which HDXmodeller is able to validate HDX-MS optimisation runs allows users adopt a sandpit approach to modelling their data, and we encourage potential users to test different input and optimisation strategies. However, after testing many different methods and input files in-house, our recommended workflow for the use of HDXmodeller is as follows. Prior to a full production run, we recommend that protein data is first split into different subsections which should then be submitted as separate jobs. Subsections may be present due to natural breaks in the HDX-MS data or created after the elimination of peptides with low occupancy or following the deletion of bridging peptides that connect two otherwise independent subsections. https://hdxsite.nms.kcl.ac.uk/Utility: A utility workspace in HDXsite contains many useful ancillary tools including a $k_{ch}$ calculator (k-intrinsic), an $\hat{R}$-matrix evaluator for customised domains (R-evaluator) and a tool for determining peptide occupancy (Occupier), which can be instructive in the preparation of data subsections. It is difficult to provide a general guide for the preparation of data subsections. However, since each subsection yields a unique validation score we recommend that users attempt to maximise data subdivision where possible as this will best enable localised validation feedback throughout input data. Nevertheless, users should also refrain from needless peptide deletion in the preparation of data subsections as this may negatively impact the level of constraint and the accuracy of model outputs (Fig. 4a). Our recommendation is to test different input strategies utilising the auto-validation outputs as a guide. After the preparation of input files, we recommend that data are first subjected to a short evaluation step involving optimisation for 10 replicates with default settings. Users will also need to upload a separate $k_{ch}$ file for these calculations which can be prepared from sequence data within our HDXutilities workspace. Following this evaluation step the $\hat{R}$-matrix score of each dataset should be inspected for guidance on anticipated modelling outcomes at the production phase. In-house benchmarking has

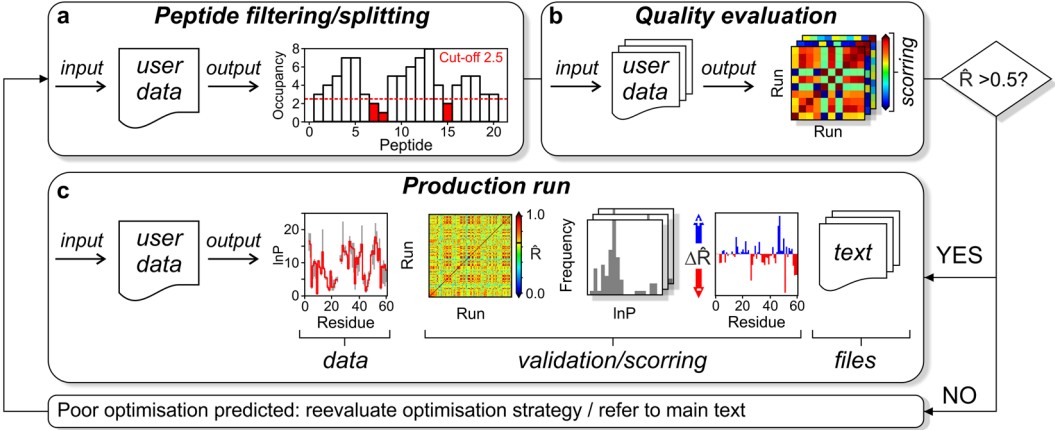

**Fig. 4 Recommended workflow for HDXmodeller. a** Input files should be in the form of RFU for each isotopic labelling timepoint and peptide that have been corrected for back and forward exchange. To increase the resolution to which optimisation outputs can be validated it is recommended for whole datasets to be split into different subsections and each subsection submitted as a separate job. Peptide filtering should also be performed using the peptide occupancy tool to identify poorly constrained peptides which we recommend eliminating from input files. The deletion of weak peptides with poor occupancy scores can also assist in the preparation of data subsections. **b** A quality evaluation step is recommended involving 10 replicate calculations using default settings. Following the evaluation stage, the R̂-matrix value of each output should be inspected for guidance relating to anticipated accuracy at the production phase. Production runs for data that obtain R̂-matrix values <0.5 at the evaluation stage is not recommended (refer to main text). **c** Production runs entail data optimisation using 50 replicates and users should inspect all outputs once the calculations are complete. HDXmodeller prepares several data outputs including various text files as well as plots of the lnP of each residue along with the interquartile range. A range of plots for data validation and scoring are also prepared, including the correlation matrix for the whole subsection as well as optimisation histograms and the ΔR̂-matrix plot for all residues.

shown 10 replicates to be sufficient to predict overall accuracy for production runs with evaluation and production R̂-matrix values typically occupying the same bins, high (>0.7), fair (0.69-0.5), or low (<0.5) with regard to data accuracy (Supplementary Fig. 4). Data with an R̂-matrix >0.5 at the evaluation phase should be submitted to a full production run whereas data <0.5 at this stage is predicted to have poor optimisation outcomes and should not be submitted. For data <0.5 further filtering of peptides may be required, and we also recommend that users go back to their original HDX-MS datasets to search for additional peptides in subsections that yield poor evaluation scores (Fig. 4b).

Additional options for data that is predicted to optimise poorly at the production phase is to merge different subsections into a single file and submit the combined data as one input. Some variations in modelling outcomes can be expected depending on whether subsections are submitted separately or merged and submitted as a single file. Following the data evaluation or production run, information regarding the quality of the optimisation for each subsection in the combined input can be gained using our stand-alone R-evaluator tool, located in the HDXutilities workspace. This tool can perform all matrix calculations on users-defined regions of a single file with no loss in accuracy provided that the evaluated regions represent discrete subsections (Supplementary Fig. 5). The matrix evaluator tool has additional applications, and users should be alerted to residues in either the evaluation or production stages with high lnP values that also display exceptionally broad inter-quartile ranges spanning 20–30 lnP. This may be indicative of data collapse for these residues and can occur in particular if the dataset contains insufficient RFU sampling at long timepoints and is a consequence of having a weak constraint on the lower optimisation bound (Methods). Ideally the longest isotope labelling timepoint should achieve an RFU as close to 1.0 as possible. This effect can generally be remedied if data subsections are combined and submitted as a single file with R̂-matrix calculated manually post-optimisation using the stand-alone evaluator tool. If the submission of a merged file does not rescue this feature we recommend that

amino acids displaying these characteristic are ignored. A final resort for data which is predicted to have a poor optimisation outcome is for users to continue to the production phase but only interpret lnP with positive ΔR̂-matrix values. However, for data with poor scores our main recommendation is for the acquisition of additional experimental data.

For HDXmodeller production runs we recommend 50 replicates with default settings. Optimisations performed with >50 replicates or >1000 maximum iterations are unnecessary, and all in-house tests performed have reported no gains in accuracy for optimisations that exceed these values. Following optimisation, users will receive a compressed file containing their outputs comprising of various raw data files and interactive graphs summarising their results. Users will receive plots of the residue lnP with projected inter-quartile range along with different validation graphs including separate optimisation histograms for all residues, the auto-validation matrix and R̂-matrix score and a plot of the ΔR̂-matrix values for each amino acid. The R̂-matrix score should first be inspected as a guide to the overall accuracy of the lnP across a data subsection, followed by examination of the optimisation histograms and ΔR̂-matrix values for information regarding accuracy on the residue level. For datasets that achieve high R̂-matrix scores (>0.7) in-house benchmarking has reported a high abundance (60–85%) of highly accurate lnP that are within ±1.0 lnP of the true values. While a reduction in accuracy is expected for data with R̂-matrix scores <0.5 many of these outputs have been shown to contain >50% highly accurate lnP following in-house testing. Insight into the accuracy of model outputs at the residue level should first be understood by inspection of the optimisation histograms of all residues. Subsidiary information regarding residue resolved accuracy can also be obtained from examination of the ΔR̂-matrix values for each residue (Fig. 4c).

Our motivation for the development of HDXmodeller was to provide a user-friendly fully automated online platform allowing high-resolution characterisation of proteins by HDX-MS. It is our aim to restore the fine details of HDX that are unfortunately lost

as a consequence of characterisation by mass spectrometry. The most prominent feature of HDXmodeller is its capacity for post-optimisation validation which provides users with important feedback on the anticipated accuracy of modelled outputs. The auto-validation feature uses a covariance matrix to compare pair-wise optimisation replicates and from this HDXmodeller can quantify the degree to which input data is constrained considering the peptide map, RFU distribution, and the underlying exchange rates. We propose that the auto-validation matrix is an image of the error surface confronted during optimisation and that high R̂-matrix scores are indicative of optimisations where the global minimum can be found more readily. We expect continued development of HDXmodeller to increase overall performance and model accuracy as well as enhance the capacity to assign confidence to the exchange rates of individual residues. The overarching objective is for HDXmodeller to become a routine tool for all HDX-MS post-processing workflows allowing the maximum possible understanding of protein function by HDX-MS.

## Methods

**Preparation of reference HDX-MS data.** Reference HDX-MS data were prepared using previously obtained HDX-MS peptides maps of alpha lactalbumin, barnase, enolase, and serum amyloid P component with lnP simulated according to previously described methods[22]. To prepare reference HDX-MS data lnP values were first simulated from protein structures according to an in-house version of a well-known expression of protein HDX behaviour[3,23]. The simulated lnP were then used to calculated $k_{obs}$ for each residue allowing the RFU of each peptide to be determined according to the following polyexponential function where, $n$ and $t$ represent number of amino acids and experimental time point, respectively (Eq. (3)).

$$RFU = \frac{1}{n}\sum_{n=1}^{n} 1 - exp(-k_{obs}.t) \qquad (3)$$

RFU values were projected at 0.25, 1, 5, 20, 60, 240, and 480 min with proline residues discounted along with amino-terminal groups of each peptide. Additional reference datasets were also prepared from the alpha lactalbumin and barnase peptide maps but using alternate lnP obtained from fitting of experimental data. During the development of HDXmodeller, many different RFU reference files were prepared and tested but it is not possible to report on all outputs. All six protein datasets for the reference library were then subjected to filtering involving the deletion of peptides with occupancy scores <2.5 (Eq. (2)). Data were then inspected and where possible independent subsections prepared, generating a library of 30 independent HDX-MS RFU data for which the underlying lnP of each residue was known. Simulated HDX-MS data was then submitted for optimisation by HDXmodeller and the with the success of the optimisation gauged by direct comparison of the simulated and modelled lnP. To investigate the role of peptide maps on optimisation outcomes 3 different lnP datasets were simulated using an identical peptide map taken from residues 309–322 of enolase. For dataset 2 the lnP were inverted whereas for dataset 3 the lnP values were randomised. RFU were then projected onto the peptide maps generating 3 unique RFU HDX-MS data which were then submitted for optimisation by HDXmodeller (Supplementary Fig. 3).

**HDXmodeller.** HDXmodeller is written in Python using standard Python libraries including Scipy, Matplotlib, scikit-learn, and Numpy. The program reads experimental HDX-MS data comprising of peptide RFU values for user-defined time-points, proline residues, and the amino-terminus are ignored. HDXmodeller also reads an additional input file that lists the chemical exchange rate constants ($k_{ch}$) for each residue. This information can be provided by users or input files can be generated online within our HDXutilities section with users simply providing the amino acid sequence, pD, and temperature using previously reported near neighbour correction terms[21,24]. HDXmodeller applies Sequential Quadratic Programming (SQP) as a successful nonlinear optimisation method which is able to handle different constraints and bounds[25]. The SQP methodology is formed as shown in Eq. (4), where $f(x)$ is the objective function, "$R$" is a real number and $h(x)$ presents the equality constants. Bound constraints are defined for $k_{obs}$ with the upper bound of each residue equal to the $k_{int}$ value.

$$\begin{aligned} &\text{minimise } f(x)\\ &\text{over } x \in R\\ &\text{subject to } h(x) = 0\\ &\text{bound constraints } 10^{-18} < x < k_{int} \end{aligned} \qquad (4)$$

HDXmodeller first generates an initial guess for the $k_{obs}$ value of each amino acid. The RFU value of each peptide is then modelled and $k_{obs}$ is optimised based on the mean squared error (MSE) between the experimental and model RFU. Both

$f(x)$ and $h(x)$ consider the MSE of the modelled and experimental RFU and these calculations are performed for every single iteration during the optimisation as shown in Eq. (4), where $x$ represents the RFU value for each peptide and $k$ is the number of the last peptide.

$$MSE\Big([x_1, x_2, x_3, \cdots, x_k]_{\text{Mod.}}, [x_1, x_2, x_3, \cdots, x_k]_{\text{Exp.}}\Big) \qquad (5)$$

All the RFU values for different time points are considered in the calculation as shown in Eq. (3). The default maximum iterations and the precision goal for termination of the optimisation are set to 1000 and $10^{-6}$, respectively. The initial guess values for $k_{obs}$ of each residue are generated from an exponential distribution scaled to achieve optimum results and using different randomly generated seed numbers for each replicate. HDXmodeller is capable of functioning with large input files and is able to analyse all of the input data and optimise $k_{obs}$ values based on the applied bounds and constraints without any loss of performance. Bounds for $k_{obs}$ are considered for each residue and are represented by $k_{ch}$ as a maximum bound and a minimum bound of $10^{-18}$ min$^{-1}$ which facilitates good optimisation speeds whilst providing a weak constraint for systems with extremely high protection factors. Optimisation default settings are set to 50 replicates following which all data is combined and various output images summarising the results of the optimisation are prepared, including a plot of the median lnP of each residue and the interquartile range, individual histograms showing the lnP density for each residue, the auto-validation matrix that quantifies the overall accuracy of the optimisation and the ΔR for each residue. Several text files are also generated including a summary of all lnP values and RFU values of all peptide models for each optimisation run.

**Auto-validation matrix.** The validation score (R̂-matrix) is calculated in the terms of pairwise correlation coefficients (R) between the lnP values of replicate runs and is based on a covariance matrix (C) as defined in Eq. (6) where, R and C represent the respective correlation coefficient and covariance matrix between replicates $i$ and $j$, and Eq. (7) where PF and M are the metrices including model lnP values ($r$) for all replicate runs and pair-wise correlation coefficient (R) values, respectively.

$$R_{ij} = \frac{C_{ij}}{\sqrt{C_{ii} \cdot C_{jj}}} \qquad (6)$$

$$PF = \begin{pmatrix} [r_1 r_2 r_3 \cdots r_n]_1 \\ [r_1 r_2 r_3 \cdots r_n]_2 \\ [r_1 r_2 r_3 \cdots r_n]_3 \\ \vdots \\ [r_1 r_2 r_3 \cdots r_n]_m \end{pmatrix} \quad M = \begin{pmatrix} R_{(1,1)} & R_{(1,2)} & R_{(1,3)} & \cdots & R_{(1,m)} \\ R_{(2,1)} & R_{(2,2)} & R_{(2,3)} & \cdots & R_{(2,m)} \\ R_{(3,1)} & R_{(3,2)} & R_{(3,3)} & \cdots & R_{(3,m)} \\ \vdots & \vdots & \vdots & \vdots & \vdots \\ R_{(m,1)} & R_{(m,2)} & R_{(m,3)} & \cdots & R_{(m,m)} \end{pmatrix} \qquad (7)$$

Following pairwise analysis R̂-matrix is reported as the arithmetic mean of all R values in the matrix (Eq. (8)), where $R_n$ is the correlation coefficient between different replicate runs in the matrix and $j$ is the total number of replicate runs.

$$\hat{R}\text{-matrix} = \sum_{n=1}^{j.j} R_n \qquad (8)$$

The accuracy of the auto-validation matrix was investigated using the library of reference data constituting 30 different HDX-MS subsections. Each subsection was submitted to HDXmodeller for optimisation and, once complete, the R̂-matrix score for each subsection was calculated and noted. The correlation coefficient between the lnP of each replicate and the reference lnP was then obtained additional R values from with the arithmetic mean was taken to obtain an R̂-reference score for each optimisation. The accuracy of the auto-validation matrix was confirmed by direct comparison of the R̂-matrix and R̂-reference scores across all data (R² 0.99). The auto-validation matrix and R̂-matrix score is automatically prepared by HDXmodeller following optimisation. Matrix calculations can also be performed on user-defined subregions of optimised data using the output lnP text file of all replicates. This standalone feature can be access in the HDXutilities workspace using the R-evaluator tool and can be performed on any region of data that constitutes an independent peptide subsection without any loss in performance.

**Validating lnP for individual residues using ΔR̂-matrix scores.** The R̂-matrix scores can also be used to estimate the accuracy of the modelled lnP for individual amino acids. These scores are calculated using identical operations to achieve R̂-matrix for a whole dataset but sequentially omit each amino acid in turn from the calculations to derive the change in R̂-matrix or the ΔR̂-matrix score of each residue. This is based on knowledge that the removal of outlying data will increase the overall correlation coefficient resulting in a negative ΔR̂-matrix value whereas the opposite should be true for more accurate lnP values. This approach was validated in several ways, the first of which was to compare the ΔR-matrix score of each amino acid with an equivalent score based the correlation coefficient between the lnP of each replicate run and the reference lnP (ΔR̂-reference). Over the 30 reference HDX-MS dataset good equivalence was shown between the direction of the ΔR̂-matrix and ΔR̂-reference scores. To understand this further an additional validation method was used involving a binary classification test on the direction of

the $\Delta\hat{R}$-matrix scores. The $\Delta\hat{R}$-matrix direction was calculated for all residues in the reference library accounting for almost 800 amino acids. For each residue the lnP RMSE was also calculated using the reference lnP values from which each dataset was constructed. A receiver operating classification (ROC) curve was then prepared which yielded an AUC value of 0.72 indicating that the error in the lnP of each residue could correctly classify the direction of $\Delta\hat{R}$-matrix to an accuracy of 72%. Data were also pooled for amino acids depending on the direction of their $\Delta\hat{R}$-matrix scores and histograms prepared for the RMSE error with bin sizes of 1 lnP. The histograms demonstrated that highly accurate model lnP with RMSE < ±1.0 were 60% more frequent for residues with positive $\Delta\hat{R}$-matrix whereas outlier amino acids with lnP > ±2.0 were 300% more likely in residues for which $\Delta\hat{R}$-matrix was negative.

**Statistics and reproducibility**. ROC plot statistics were calculated using embedded macros in SigmaPlot v14 (Systat Software Inc. USA) and default settings. Data were ranked according to their RMSE and classified as either positive or negative depending on the direction of the $\Delta\hat{R}$-matrix.

**Reporting summary**. Further information on research design is available in the Nature Research Reporting Summary linked to this article.

## Data availability
All data used in this study is available to download https://hdxsite.nms.kcl.ac.uk/.

## Code availability
All codes may be accessed for use as is from our webserver https://hdxsite.nms.kcl.ac.uk/.

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

## Acknowledgements
This work was funded by the Biotechnology and Biological Sciences Research Council (BBSRC). We also gratefully acknowledge the use of the research computing facility at King's College London, Rosalind (https://rosalind.kcl.ac.uk). We also thank Andy McAllister for design of the website logos.

## Author contributions
R.E.S. and A.J.B. designed the experiments, protocols, analysed data, and wrote the manuscript. R.E.S. developed and optimised the code and the website.

## Competing interests
The authors have no competing interests.
