## [Peer Review File · Communications Biology]

Reviewers' comments:

Reviewer #2 (Remarks to the Author):

SUMMARY

HDX-MS is a technique that probes the secondary and tertiary structures of proteins as quantified by amide protection factor (PF). A major limitation of the technique is that measurements are limited to the resolution of peptides, typically 5-20 residues in length, unlike the NMR version of this experiment where the PFs of individual backbone amides may be obtained. The present work describes a new algorithm and accompanying webserver that seeks to solve this longstanding problem by attempting to modulate the individual PF values using numerical methods based on measured HDX kinetics of many overlapping peptides. Achievement of this goal would be highly prized in the HDX-MS community. Thus, this work is of high potential interest within the HDX-MS subfield. Importantly, the authors recognize that the problem is under-determined and they have developed useful metrics for evaluating the quality of the estimated PF values that are obtained.

1. Given the importance of this problem in the HDX-MS community (almost a Holy Grail, in fact), this is not the only algorithm that attempts to solve this problem. Regrettably, the manuscript makes no mention of a rather large body of literature describing similar approaches to solving this problem. A partial list of those citations follows:

- Babic, D.; Kazazic, S.; Smith, D. M., Resolution of protein hydrogen/deuterium exchange by fitting amide exchange probabilities to the peptide isotopic envelopes. *Rapid Commun Mass Spectrom* 2019, 33 (15), 1248-1257.
- Babić, D.; Smith, D. M., Localization improvement of deuterium uptake in hydrogen/deuterium exchange in proteins. *J. Chemometrics* 2017, 31 (3), e2876.
- Althaus, E.; Canzar, S.; Ehrler, C.; Emmett, M. R.; Karrenbauer, A.; Marshall, A. G.; Meyer-Base, A.; Tipton, J. D.; Zhang, H. M., Computing H/D-exchange rates of single residues from data of proteolytic fragments. *BMC Bioinformatics* 2010, 11, 424.
- Zhang, Z.; Zhang, A.; Xiao, G., Improved protein hydrogen/deuterium exchange mass spectrometry platform with fully automated data processing. *Anal. Chem.* 2012, 84 (11), 4942-9.
- Zhang, Z., Complete Extraction of Protein Dynamics Information in Hydrogen/Deuterium Exchange Mass Spectrometry Data. *Anal. Chem.* 2020.
- Hamuro, Y.; Zhang, T., High-Resolution HDX-MS of Cytochrome c Using Pepsin/Fungal Protease Type XIII Mixed Bed Column. *J. Am. Soc. Mass Spectrom.* 2019, 30 (2), 227-234.
- Kan, Z. Y.; Walters, B. T.; Mayne, L.; Englander, S. W., Protein hydrogen exchange at residue resolution by proteolytic fragmentation mass spectrometry analysis. *Proc Natl Acad Sci U S A* 2013, 110 (41), 16438-43.
- Kan, Z.-y.; Ye, X.; Skinner, J. J.; Mayne, L.; Englander, S. W., ExMS2: An Integrated Solution for Hydrogen-Deuterium Exchange Mass Spectrometry Data Analysis. *Anal. Chem.* 2019, 91 (11), 7474-7481.
- Liu, T.; Pantazatos, D.; Li, S.; Hamuro, Y.; Hilser, V.; Woods, V., Quantitative Assessment of Protein Structural Models by Comparison of H/D Exchange MS Data with Exchange Behavior Accurately Predicted by DXCOREX. *J. Am. Soc. Mass Spectrom.* 2012, 23 (1), 1-14.
- Claesen, J.; Politis, A., POPPeT: a New Method to Predict the Protection Factor of Backbone Amide Hydrogens. *J. Am. Soc. Mass Spectrom.* 2019, 30 (1), 67-76.

2. It is important for a revised manuscript to cite this literature and more clearly define what is most useful in the present approach. It appears to me that the novel aspects of HDX-Modeller are the specific optimization strategy, its autovalidation, and that a webserver is available.

3. The structure of the manuscript could be improved. Most of the theory, where the novelty lies, is relegated to the online methods. The discussion is really just a tutorial of best practices for using the software. More deeply explicating how to interpret autovalidation would be useful in the discussion.

4. The authors have only validated their algorithm with simulated experimental data. This was an unfortunate choice. Residue-resolved PFs and peptide maps for a number of proteins are available (see works by Kan and Hamuro, for example). The work would benefit from an analysis of true experimental data. Additionally, the clarity would be improved if "simulated" were prepended to

the phrase "experimental data" where appropriate.

5.No details are provided about how the simulated data were generated or how diverse these data sets were.

6.PF values seem to vary somewhat smoothly from residue to residue. How does the algorithm perform when the PF values vary more sharply along the length of the protein?

7.Englander recently updated the reference values for chemical exchange (misnamed as "intrinsic exchange" in this manuscript), see J. Am. Soc. Mass Spectrom. 2018, 29 (9), 1936-1939. The authors may need to adjust their reference values somewhat.

Reviewer #3 (Remarks to the Author):

The manuscript by Salmas and Borysik reports on a potentially valuable resource for investigators employing HDX-MS to study protein structure and dynamics. The methods and their strengths and weaknesses are clearly described, and the associated webserver will likely prove useful to the HDX-MS community. I have only minor comments.

1) At several points, the authors recommend breaking experimental HDX-MS data sets into subsections to submit for analysis. Some additional information on this would be helpful. What size subsets are recommended? Should peptide data sets be combined from contiguous regions of the protein or based on some other criterion? How many distinct subsets are recommended?

2) While the authors note several existing methods for extracting single residue PFs from peptide level data sets, the method of Saltzberg et. al. (J Phys Chem B. 2017 Apr 20; 121(15): 3493-3501) was not mentioned. This should be included in the list of existing methods.

3) Speaking of existing methods, at least for Saltzberg and for Skinner et. al. (reference 12) the code implementing their methods is publicly available. Is there a reason the authors did not perform a direct comparison between their method and these existing ones?

Answers to reviewer comments

Reviewer 1

Q) It is unclear to me how the experimental standard deviations for the peptides transfer into the algorithm's output for standard deviations for the residue's deuterium uptake or how the experimental deviations affect the output of the algorithm.

A) We have looked into this and found no relationship between input errors and errors in the fit. We tried taking the reciprocal of the input SD to rank peptides during optimisation but saw no benefit from this to the approach. I think this is something worth looking into but probably in a standalone paper where it can be opened up.

Comments

- 1) Line 8: HDX benefits from low protein size restrictions, meaning should be
- 2) Line 205-207: Should be reworded, it would benefit from being flipped 180°

A) We have addressed both of these comments please refer to manuscript with track changes.

Reviewer 2

Q) Given the importance of this problem in the HDX-MS community (almost a Holy Grail, in fact), this is not the only algorithm that attempts to solve this problem. Regrettably, the manuscript makes no mention of a rather large body of literature describing similar approaches to solving this problem. It is important for a revised manuscript to cite this literature (1) and more clearly define what is most useful in the present approach (2). It appears to me that the novel aspects of HDX-Modeller are the specific optimization strategy, its autovalidation, and that a webserver is available.

A1) We have now included many of the papers listed by the reviewer except for the article by Hamuro (2019) as this method only uses peptide subtraction to obtain rates from overhangs and doesn't really entail any optimisation and the papers from Liu (2012) and Classen (2019) because they're entirely unrelated to this work and are to do with simulating PFs from structures rather than modelling them from HDX-MS data. The paper requested by Althaus (2010) was originally included in the article. The reviewer is correct in that the body of literature is large so rather than include everything we originally decided to omit papers that did not include any method validation, Zhang for example. The papers from Babic and Kan are slightly different in that they attempt to model the ion envelopes. This is a good idea but is risky as it requires a sizeable assumption that all peptides are random coils under quench conditions so that back exchange can be modelled at the envelope level, i.e. residue resolution. Accordingly, we originally left these papers out, but we have now included them as requested.

A2) We think the paragraph in the introduction already spells out the novelty of HDXmodeller. We would rather not add to this and draw out the text unnecessarily *"Here we report HDXmodeller the world's first online webserver for high-resolution HDX-MS. HDXmodeller is a fully automated advanced programming tool capable of calculating residue resolved HDX protection factors (PFs) from peptide level HDX-MS input data. Through an extensive search of different algorithms and procedures, HDXmodeller is able to provide the most accurate high-resolution exchange rate models currently reported, depending on the input. The standout feature of HDXmodeller however, is an auto-*

validation function that takes into consideration the quality of the entire optimisation process through the use of a novel method based on a covariance matrix over different replicates. Crucially, the auto-validation feature can quantify the fidelity of the model output to an accuracy of 99% without prior knowledge of the underlying residue exchange rates. HDXmodeller will provide the growing number of HDX-MS practitioners easy access to high-resolution HDX-MS along with essential insight into the reliability of their data.”

Q) The structure of the manuscript could be improved. Most of the theory, where the novelty lies, is relegated to the online methods. The discussion is really just a tutorial of best practices for using the software. More deeply explicating how to interpret autovalidation would be useful in the discussion.

A) We have purposefully located all of the more cumbersome theory in the online methods because this arrangement aligns with our strategy to make these methods accessible and easy to use/digest. Accordingly, we think that space in the discussion is best served by outlining best strategies to encourage use of the webserver, with the theory provided elsewhere. However, we have extended the discussion to provide some more details of the validation matrices, please see article with track changes.

Q) The authors have only validated their algorithm with simulated experimental data. This was an unfortunate choice. Residue-resolved PFs and peptide maps for a number of proteins are available (see works by Kan and Hamuro, for example). The work would benefit from an analysis of true experimental data.

A) We realise now that we failed to provide sufficient information on how the reference datasets were prepared nor have we provided any justification behind our approach which was settled on after much deliberation. We did consider using available NMR derived protection factors for the development of HDXmodeller, but we had several concerns which ultimately guided us to our present approach.

There are actually very few NMR datasets for proteins in non-denaturing conditions and most are only around 30 – 50% complete. These data are also very old and were reported throughout the 1990s when the technique was most popular. This is not a problem in itself, but the authors do not report the uncertainty in the protection factors, nor do they demonstrate the fit from which these values were derived for anything but a few, and probably the most successful, amino acids. The use of this data therefore introduces risks which could hinder the development of a robust modelling method. Many of these older datasets were also obtained prior to the development of CLEANEX-PM which enabled fast-HSQC acquisitions. This adds to their incompleteness as NMR was incapable of measuring the exchange rates of the least protected amino acids. In general, most of these papers report protection factors for a narrow band of amino acids with mid-range protection and are therefore unsuitable for method development. A commitment to an NMR-based approach would also require the use of experimental HDX-MS data as it would not be possible to simulate data from the incomplete NMR protection factors. This adds still further ambiguity into method development owing to potential errors in the experimental HDX-MS such as those introduced from poorly implemented correction of extraneous exchange. The HDX-MS data from Hamuro, for example, cannot be used by us as they don't appear to account for back exchange correction at all in their paper. There is definitely a need to revisit these HDX-NMR and HDX-MS comparisons, but it needs to be approached properly and I'd be extremely cautious of overinterpreting results from different datasets without any knowledge of the uncertainty.

At present we think these methods are best served using a theoretical approach with benchmarking performed using data simulated with our current best understanding of HDX-MS principles. For us this approach deals with a large proportion of the uncertainty, but it raised the important question of how best to prepare the simulated data. Looking at the literature several authors, including Zhang and Saltsburg do not appear to have provided any benchmarking of their methods at all. Others who have opted for an approach involving simulations have prepared data to test their method which is entirely synthetic, and which is not based on any available theories. For example, in the paper by Skinner, benchmarking was performed using synthetically smooth data which would make projections with real data difficult to ascertain (below).

To this end we simulated all of our $\ln P$ values from protein crystal structures using a well-established algorithm of simulating these data first proposed by Vendruscolo. This algorithm was benchmarked against available NMR data and although there are no current expressions which can simulate protection factors accurately they should at least maintain their character and crucially for us the values are known.

We had some concerns regarding the performance of our approach with real HDX-MS data mainly due to its lack of self-consistency as compared with simulations which are perfectly smooth. We are currently working on a follow up article focussing on several areas including error handling in the modelling of experimental data. If we compare the modelling outputs obtained from experimental HDX-MS data against a simulated profile the correlation between the two is extremely high. We have also performed many additional comparisons and the performance of the auto-validation method in terms of the R-matrix value does not appear to be impeded when we apply our approach to experimental data (below).

At present we think that our approach is valid and entirely reasonable for method development / validation. We do however accept that the explanation / justification of our approach is inadequate, and we have therefore expanded considerably the short section in the results that relates to this, please see article with track changes.

Q) No details are provided about how the simulated data were generated or how diverse these data sets were.

A) We have added a new figure in the SI which demonstrates the method of preparing simulated data. We have also provided all of the data that was used in the preparation of this article which can now be downloaded as a zip file from our webserver using the “example” link.

Q) PF values seem to vary somewhat smoothly from residue to residue. How does the algorithm perform when the PF values vary more sharply along the length of the protein?

A) We are not sure exactly what is meant by this. The reference PFs have been simulated from protein structures using a well-known method and exhibit considerable variation between amino acids. Perhaps the PFs appear smooth in some of the figures because the x-axis is drawn out considerably relative to the y-axis? The figure below displays the data from figure 1a with a square figure frame and the reference data are highly variable, particularly over the first 60 amino acids.

We don't think protection factors are expected vary wildly between residues and, although there will be exceptions, they reflect the different domains the amino acid chain must thread through such as the slow exchange core. Most optimisation methods will struggle with highly variable data. However, the degree of modelling success is complex and doesn't only depend on the PFs but also the labelling time points, the peptide map and the synergy between all the different constraints. The important point is to have a method that can flag any difficulties in modelling and also help users identify any problem areas.

We looked into this and took one reference dataset (residues 79 – 124 from the figure above) and increased the variation between the PFs by introducing different degrees of Gaussian noise for each value from 1 to 3 standard deviations. The results are shown below in 4 columns from 0 to 3 SD. The top row shows the relationship between the reference and modelled lnP, also shown in the scatter plot directly below this. The 3rd row is a histogram of the lnP RMSE of all replicate runs – not the central tendency. The 4th row shows the ΔR -matrix values which is part of our validation output and which can be used to help differentiate accurate outputs (positive values) from inaccurate outputs (negative values) of each amino acid. The 4th row shows the auto-correlation matrices for each input and the associated R-matrix score which is a summary of the anticipated accuracy of each optimisation. The single plot at the bottom of the figure shows the alignment of these new R-matrix scores with our calibration dataset.

While we don't think this is particularly realistic there is much more variation in the reference lnP as we increase the SD. Although some outliers are introduced our method handles the increased variation very well. Even at the highest SD ~ 60% of the modelled lnP are within ± 1 of the reference values. The important thing to note is that our method is able to detect problems with the modelling. This is demonstrated by the many shaded boxes in plots on the top row, which wrap around particularly problematic regions, which we have also mirrored in the ΔR -matrix plots on row 4. We can see that these difficult regions almost unanimously yield negative scores so that our validation method is able to successfully flag these regions. The auto-correlation matrices are similar for all inputs, except column 2 (1SD) which is also the poorest fitting dataset. These new R-matrix scores are also in line with what we expect from these data as can be seen from the figure on the bottom. We are confident that our method can handle a range of input data and crucially it is able to identify any problems with the modelling should they arise.

Q) Englander recently updated the reference values for chemical exchange (misnamed as “intrinsic exchange” in this manuscript), see J. Am. Soc. Mass Spectrom. 2018, 29 (9), 1936-1939. The authors may need to adjust their reference values somewhat.

A) Our database of correction terms already includes these new values; we have made this clearer in the Methods. We have changed all kint -> kch, although the terms kint, kch, kex and even krc are used somewhat interchangeably particularly between different communities.

Reviewer 3

Q) At several points, the authors recommend breaking experimental HDX-MD data sets into subsections to submit for analysis. Some additional information on this would be helpful. What size subsets are recommended? Should peptide data sets be combined from contiguous regions of the protein or based on some other criterion? How many distinct subsets are recommended?

A) This is a good question and basically the subsections should be as small as possible without compromising the optimisation due to excessive peptide deletion. The approach we recommend is for different input strategies to be tested using feedback from the auto-validation outputs as a guide trying to maximise data splitting. We have added some additional text in the discussion hopefully this will clarify this a bit better.

Q) While the authors note several existing methods for extracting single residue PFs from peptide level data sets, the method of Saltzberg et. al. (J Phys Chem B. 2017 Apr 20; 121(15): 3493–3501) was not mentioned. This should be included in the list of existing methods.

A) We have now included this citation in the introduction.

Q) Speaking of existing methods, at least for Saltzberg and for Skinner et. al. (reference 12) the code implementing their methods is publicly available. Is there a reason the authors did not perform a direct comparison between their method and these existing ones?

A) I think a cross platform comparison of different methods is something worth pursuing at some point, but this should be set up with all PIs input and with a clear objective. This has never been done previously and is not the scope of the present article.

REVIEWERS' COMMENTS:

Reviewer #2 (Remarks to the Author):

In the revision and rebuttal, the authors have either (1) adequately responded to my comments or (2) adequately explained why they have chosen not to.

I found one typographical error: p. 15: several instances of kint were not changed to kch

Reviewer #3 (Remarks to the Author):

This revised manuscript adequately addresses my previous concerns.